# Molecular Cloning, Screening of Single Nucleotide Polymorphisms, and Analysis of Growth-Associated Traits of *igf2* in Spotted Sea Bass (*Lateolabrax maculatus*)

**DOI:** 10.3390/ani13060982

**Published:** 2023-03-08

**Authors:** Sigang Fan, Pengfei Wang, Chao Zhao, Lulu Yan, Bo Zhang, Lihua Qiu

**Affiliations:** Guangdong Provincial Key Laboratory of Fishery Ecology and Environment, South China Sea Fisheries Research Institute, Chinese Academy of Fishery Sciences, Guangzhou 510300, China

**Keywords:** *Lateolabrax maculatus*, *insulin-like growth factor 2* (*igf2*), single nucleotide polymorphism (SNP), association analysis, growth trait

## Abstract

**Simple Summary:**

The spotted sea bass (*Lateolabrax maculatus*) is an economically important fish species cultured in China. Single nucleotide polymorphisms (SNPs) associated significantly with growth traits were screened to promote culturing conditions for *L. maculatus*. The insulin-like growth factor 2 gene (*igf2*) of *L. maculatus* was isolated. Fourteen SNPs were detected from the genome sequence of *igf2*. Four SNPs were associated significantly with the growth traits of *L. maculatus*. The SNPs we identified could be valuable for the genetic breeding of *L. maculatus*.

**Abstract:**

The insulin-like growth factor 2 gene (*igf2*) is thought to be a key factor that could regulate animal growth. In fish, few researchers have reported on the single nucleotide polymorphisms (SNPs) located in *igf2* and their association with growth traits. We screened the SNPs of *igf2* from the spotted sea bass (*Lateolabrax maculatus*) by Sanger sequencing and made an association between these SNPs with growth traits. The full-length complementary (c) DNA of *igf2* was 1045 bp, including an open reading frame of 648 bp. The amino acid sequence of Igf2 contained a signal peptide, an IGF domain, and an IGF2_C domain. Multiple sequence alignment showed that the IGF domain and IGF2_C domain were conserved in vertebrates. The genome sequence of *igf2* had a length of 6227 bp. Fourteen SNPs (13 in the introns and one in one of the exons) were found in the genome sequence of *igf2*. Four SNPs located in the intron were significantly associated with growth traits (*p* < 0.05). These results demonstrated that these SNPs could be candidate molecular markers for breeding programs in *L. maculatus*.

## 1. Introduction

In fish, the insulin-like growth factor (Igf) family includes three Igfs, two Igf receptors, and six Igf-binding proteins [1,2]. These proteins have important roles in regulating reproduction, metabolism, growth, and development in fish [3,4,5]. In mammals, IGF2 is an evolutionarily conserved peptide hormone with structural homology to proinsulin [6]. As an important protein hormone in the growth hormone (GH)/IGF axis, Igf2 can transport GH, promote glucose transport into muscle, and regulate the proliferation, differentiation, and survival of cells in fish and mammals [3,7,8].

In 1992, Igf2 was isolated from rainbow trout (*Oncorhynchus mykiss*), the first time that it had been found in fish [9]. Then, Igf2 was detected from other fish, such as *Oreochromis mossabicus*, *Danio rerio*, *Oryzias latipes*, *Cyprinus carpio*, *Ctenopharyngodon idellus*, and hybrid yellow catfish (*Pelteobagrus fulvidraco*) [2,10,11,12,13,14]. In general, vertebrates contain one *igf2*, but some scholars have found more than two *igf2* paralogs in fish. For example, two igf2 genes (*igf2a* and *igf2b*) have been detected in zebrafish and grass carp [14,15]. The existence of more than one paralog in fish is the consequence of the whole-genome duplication experienced by teleosts [16], followed by an extra round of genome duplication that occurred in salmonids and the Cyprininae subfamily [17].

A single nucleotide polymorphism (SNP) is a germline substitution of a single nucleotide at a specific position in the genome [18]. Due to their abundance, high density, and prevalence in genomes, SNPs are useful markers in genetic research [18]. For example, SNPs are the most used molecular markers in candidate gene methods and are used widely to analyze the association with growth traits [19]. Few studies have reported the association between SNPs and growth traits in fish. For instance, in European seabass (*Dicentrarchus labrax* L.), two SNPs from *igf1* are associated with body weight and total length (TL) [8]. In Atlantic salmon (*Salmo salar*), two SNP loci in *igf1* are associated with body weight [20]. In *P. fulvidraco*, an SNP of *igf2* is associated significantly with nine growth traits [2].

The spotted sea bass (*Lateolabrax maculatus*) is one of the species from the family Moronidae (Perciformes). It is distributed from the coast of China to the Indo-China Peninsula [21]. It is adapted to a wide range of temperatures and salinity [22]. Because of its fast growth rate and valuable meat quality, *L. maculatus* has become an economically important fish in China. The annual production in China in 2021 was 199,106 tons (data from China fishery statistical yearbook of 2021). Selection of new varieties of fast-growing *L. maculatus* requires studies of growth-trait-related genes and associations with molecular markers.

In the present study, the complementary (c)DNA and DNA sequences of *igf2* in *L. maculatus* were isolated and characterized. The SNPs in *igf2* were detected by direct sequencing, and their associations with the growth traits of *L. maculatus* were analyzed. These results can be used as a guide to breed fast-growing strains of *L. maculatus*.

## 2. Materials and Methods

### 2.1. Sample Collection

*L. maculatus* individuals were caught in the East China Sea and reared in net cages for several months. During the breeding season (November–January of the following year), fish with gonadal maturation were transported to a cement pool in a fish farm (Ningde Fufa, Ningde, Fujian Province, China). Ten males and four females were cross-mated. All embryos were hatched simultaneously. More than 10,000 larvae were transported to the Zhuhai Base of the South China Sea Fisheries Research Institute (Zhuhai, Guangdong Province, China). All larvae were cultured and managed in the same pond. At the age of 9 months, *L. maculatus* individuals (average standard length (SL) = 33.34 ± 2.27 cm) were chosen randomly. Body mass (BM) was measured using a fish scale. Five body measurements (SL, body depth (BD), body width (BW), head length (HL), and total length (TL)) were measured with vernier calipers (Appendix A). These growth traits were measured and analyzed at 9 months. The fins of each individual were sampled for DNA extraction. The muscle tissues of *L. maculatus* adults were excised. All samples were collected in a sterile tube, frozen by liquid nitrogen, and maintained at –80 °C until use.

### 2.2. Extraction of Total RNA and cDNA Synthesis

Total RNA from muscle was extracted using TRIzol^®^ Reagent (Invitrogen, Carlsbad, CA, USA) following manufacturer protocols. The quality and concentration of RNA were assessed by electrophoresis on 1% agarose gels and NanoDrop™ 2000 (Thermo Scientific, Waltham, MA, USA), respectively. One µg of total RNA from each sample was used as a template for cDNA synthesis using the PrimeScript™ II 1st Strand cDNA Synthesis Kit (TaKaRa Biotechnology, Shiga, Japan). First-strand cDNA was synthesized from total RNA according to manufacturer instructions.

### 2.3. Cloning igf2 from cDNA

The full-length cDNA of *igf2* was isolated using transcriptome data of *L. maculatus* noted previously (PRJNA811720). One pair of primers (cs/ca) was used to examine the sequence accuracy (Table 1). Polymerase chain reaction (PCR) amplification assays involved the use of 20 μL of master mix (2 μL of 10× Ex Taq Buffer containing Mg^2+^ (TaKaRa Biotechnology, Shiga, Japan), 0.8 μL of each primer (10 mM), 0.2 μL of Ex Taq (5 U/mL), 1.6 μL of a dNTP mixture (2.5-mM each), 0.8 μL of cDNA, and 13.8 μL of double-distilled H_2_O).

PCR conditions were 1 cycle at 94 °C for 3 min, 35 cycles at 94 °C for 30 s, 50 °C for 30 s, 72 °C for 30 s, and a 7 min step at 72 °C. PCR fragments were purified with a SanPrep Column DNA Gel Extraction Kit (Sangon Biotechnology, Shanghai, China) and then sequenced by the Sanger’s dideoxy method at Tsingke Biological Technology (Guangzhou, China).

### 2.4. Cloning of igf2 from Genomic DNA

Total genomic DNA was isolated using the Marine Animals DNA Kit (Tiangen, Beijing, China). The quality and integrity of DNA were examined by electrophoresis on 1% agarose gels and NanoDrop 2000 (Thermo Scientific, Waltham, MA, USA), respectively. DNA was dissolved in sterile water at a concentration of 50 ng/mL and stored at –20 °C.

The genome sequence of *igf2* was obtained from the *L. maculatus* genome (CP027267.1). One pair of primers (gs and ga) were used to verify the accuracy of the sequence (Table 1). PCR was conducted in a reaction volume of 20 μL (13.8 μL of double-distilled H_2_O, 2 μL of 10× LA Taq Buffer containing Mg^2+^ (TaKaRa Biotechnology, Shiga, Japan), 0.8 μL of each primer (10 mM), 1.6 μL dNTP mixture (2.5-mM each), 0.8 μL of DNA, and 0.2 μL of LA Taq (5 U/mL)). The PCR program was 1 cycle at 94 °C for 3 min, 35 cycles at 95 °C for 30 s, 52 °C for 30 s, 72 °C for 3 min, and 1 cycle at 72 °C for 7 min. As described above, PCR products were purified and sent to Tsingke Biological Technology for sequencing.

### 2.5. Sequence Analysis

ORF Finder (www.ncbi.nlm.nih.gov/gorf/orfig.cgi/, accessed on 12 October 2022) was used to obtain the open reading frames (ORFs) and predict the coding protein sequence. Primer-BLAST (www.ncbi.nlm.nih.gov/tools/primer-blast/, accessed on 12 October 2022) was employed to assist with primer design. Exons and two introns were analyzed via a comparison between the genomic DNA sequence and cDNA sequence. The signal peptide of the deduced amino acid (aa) sequence was analyzed with SignalP (www.cbs.dtu.dk/services/SignalP/, accessed on 12 October 2022). Functional domains were predicted by the conserved domains detailed in the National Center for Biotechnology Information (www.ncbi.nlm.nih.gov/Structure/cdd/wrpsb.cgi/, accessed on 12 October 2022). The chemical and physical properties of proteins were analyzed by ProtParam (http://web.expasy.org/protparam/, accessed on 12 October 2022). Global Align (https://blast.ncbi.nlm.nih.gov/Blast.cgi/, accessed on 12 October 2022) was used to analyze the similarities of the aa sequence of Igf2. Multiplex sequence alignment was carried out using Clustal Omega (www.ebi.ac.uk/Tools/msa/clustalo/, accessed on 12 October 2022). A phylogenetic tree was constructed using MEGA 7.0 with the maximum-likelihood algorithm [23].

### 2.6. Identification of SNPs and Statistical Analyses

Twenty individual DNAs were chosen randomly for SNP isolation. Four pairs of primers (E1s/E1a, E2s/E2a, E3s/E3a, E4s/E4a) were used to produce the DNA sequence of *igf2* (Table 1). PCR fragments were purified with the DNA Gel Extraction Kit (Omega BioTek, Norcross, GA, USA), ligated into the pMD18-T vector (TaKaRa Biotechnology, Shiga, Japan), and sequenced. These sequences were analyzed to screen for potential SNPs with the SeqMan program (DNASTAR, Madison, WI, USA). If SNPs were found, then PCR amplification of the remaining 170 DNA samples was done.

An association analysis between different genotypes of an SNP and growth traits was carried out using the general linear model with SPSS 19.0 (IBM, Armonk, NY, USA). The model was:Yij = μ + Gi + eij
where Yij represents the observed value of the jth individual of genotype i, μ denotes the mean of the observed values, Gi represents the effective value of the genotype i, and eij denotes the random error [24]. The significance of differences was tested using Tukey’s multiple-range test.

## 3. Results

### 3.1. cDNA Cloning and Characterization of igf2

Based on transcriptome sequencing and validation by the primers cs and ca, we obtained the cDNA sequence of *igf2*. The full-length *igf2* cDNA (GenBank accession number: ON462263) was 1045 bp with a 5′-untranslated region (UTR) of 177 bp, 3′-UTR of 220 bp, and ORF of 648 bp (Figure 1). *igf2* encoded a protein containing 215 aa. The molecular mass was 24.60 kDa and the isoelectric point was 10.04. A putative signal peptide (1–50 aa) was detected in the N-terminal sequence, an insulin-like growth factor (IGF) domain (51–117 aa), and an IGF2_C domain (147–202 aa) (Figure 1).

### 3.2. Characterization of the Genome Sequence of igf2

The genome sequence of *igf2* was 6227 bp in length (GenBank accession number: ON462264). It contained four exons and three introns (Figure 2). The length (in bp) of exons 1, 2, 3, and 4 was 75, 151, 182, and 240, respectively. Three introns had a length (I bp) of 1027, 1574, and 1458, respectively. All intron–exon boundaries were consistent with the GT-AT rule.

### 3.3. Sequence Identity and Phylogenetic Analysis of Igf2

Global Align was employed to obtain the sequence alignment and protein sequences of Igf2 in *L*. *maculatus* and other species. High identity was found between Igf2 of *L*. *maculatus* and Igf2 of other fish. For example, the sequence of Igf2 of *L*. *maculatus* was identical to Igf2 of *Lateolabrax japonicus*, and had 93% identity with Igf2 of *Siniperca chuatsi*, *Seriola dumerili*, and *Trachinotus ovatus* (Table 2). However, Igf2 of *L*. *maculatus* showed low identity with the IGF2 sequences of higher vertebrates such as mammals: only 47% similarity with *Mus musculus* and 44% with *Homo sapiens* (Table 2). The IIGF and IGF2_C domains of IGF2 were very conserved in all species, which suggested that these two domains were important (Figure 3). The phylogenetic trees showed that mammals and fish were divided into two clades and that Igf2 of *L*. *maculatus* and Igf2 of *L. japonicus* were clustered together (Figure 4).

### 3.4. Detection of SNPs

Putative SNPs were detected after comparing *igf2* sequences from 20 *L*. *maculatus* individuals. In the present study, a putative SNP locus was considered to be a nucleotide site with a variation of a base in the DNA sequence in ≥5 individuals. SNPs were detected only in sequences amplified by E3s/E3a and E4s/E4a primers. Subsequently, E3s/E3a and E4s/E4a primers were amplified further and sequenced directly, and SNP sites were detected in the remaining 170 individuals. Sequencing failures were removed. Analyses of sequence alignment showed 14 SNPs, of which 9 were transformations (A/G, C/T) and 5 were transpositions (A/C, T/G) (Table 3). The first base of the DNA sequence of *igf2* was named as the starting position of the SNP locus. Intron 2 had 10 SNPs and intron 3 had 3 SNPs (Table 3). Another SNP located in exon 4 was a non-synonymous mutation, and the encoded aa was mutated from isoleucine to valine (Figure 1, Table 3). The genotypes, numbers, and genotype frequencies of all SNP loci are listed in Table 3. The sequencing peaks of some SNPs are shown in Figure 5.

### 3.5. Association of SNPs with Growth Traits

A general linear model was employed to analyze the relationship between fourteen SNP loci and six growth traits: four SNP loci were correlated significantly with growth traits (*p* < 0.05) (Table 4). SNP g2907C>T was associated significantly with HL (*p* = 0.044) and BD (*p* = 0.035). HL and BD in the TC genotype were significantly higher than those of the CC genotype. The length of the SNP g3230A>C locus was associated significantly with TL (*p* = 0.011), and that of homozygous (AA, CC) was significantly higher than that of heterozygous (AC). SNP g3294C>T genotypes were correlated with BW (*p* = 0.002), BD (*p* = 0.044), and HL (*p* = 0.008). Genotype TT was significantly higher than genotype CC in terms of BD, BW, and HL. SNP g5064C>T genotypes were correlated significantly with individual SL (*p* = 0.027). SL of genotype CC was significantly higher than that of genotype TT.

## 4. Discussion

We cloned the cDNA of *igf2* from *L. maculatus*. Similar to other vertebrates such as mammals, birds, amphibians, and fish, the protein sequence of Igf2 of *L. maculatus* also includes an IGF domain and IGF2_C domain (www.ncbi.nlm.nih.gov/gene/3481/ortholog/?scope=7776&term=IGF2, accessed on 18 July 2022). The protein sequence of Igf2 of *L. maculatus* was found to be highly similar to that of other vertebrates, especially the two domains IIGF and IGF2_C, which were highly conserved (Figure 3). The genome structure of *igf2* of *L. maculatus* comprised four exons and three introns (Figure 2). This structure is consistent with that of other fish types, including largemouth bass (*Micropterus salmoides*), common carp (*Cyprinus carpio*), salmon (*Oncorhynchus keta*), tilapia (*Oreochromis niloticus*), Asian sea bass (*Lates calcarifer*), and pike perch (*Sander lucioperca*) [25,26,27,28,29,30,31]. This result indicates that the structure of the *igf2* genome of fish is relatively conserved. In mammals, the number of *igf2* exons is greater than that of fish. For example, in mice (*Mus musculus*), *igf2* has eight exons, and in humans and cows (*Bos taurus*) *igf2* has ten exons, respectively [32,33]. Hence, the structure of the *igf2* genome has changed considerably during evolution.

Growth traits are important for fish breeding [2]. Screening the molecular markers closely related to growth traits for direct breeding can avoid the limitations of inaccurate environmental and phenotypic measurement and improve the success rate of breeding [34]. *igf2* is a candidate gene for the selection of growth traits [2,8]. Some SNPs associated with growth-related traits have been screened and detected in *IGF2* of pigs and cattle [33,35,36]. There have also been reports on the association between *igf2* and growth traits in fish. For example, one SNP of *igf2* was found to be correlated significantly with nine growth traits of the hybrid yellow catfish (*P. fulvidraco* (♀) × *Pelteobagrus vachelli* (♂)) [2]. One SNP in intron 3 of *igf2* of *S. lucioperca* is closely related to body weight [31]. One SNP locus in exon 3 of *igf2* in tilapia (*O. niloticus*) is correlated with growth traits [29]. Xie et al. [37] demonstrated that the mutation site in *igf2* of a new strain of Yellow River carp reduced the body weight of fish. Li et al. [25] detected four SNPs in *igf2* of *M. salmoides*, and the two haplotypes composed of these loci were closely related to body weight (*p < 0.05*). In the present study, 14 SNPs were isolated from the *igf2* genome sequence of *L. maculatus*, 13 of which were located in the intron. There are many SNPs in an intron because an intron does not participate in encoding proteins, has little selection pressure, and can accumulate mutations readily [38]. In addition, SNPs in the non-coding region may have an impact on alternative splicing, splicing efficiency, mRNA degradation, and gene expression [39,40]. Similar to other studies [2,29,36], we found that four SNPs of *igf2* were associated significantly with growth traits (*p < 0.05*) (Table 4). These results indicate that: (i) *igf2* is a key potential gene for growth traits in fish; (ii) the SNPs associated with growth-related traits in *igf2* can be used as markers in studies on *L. maculatus*.

## 5. Conclusions

The cDNA and DNA sequences of *igf2* were isolated. Fourteen SNPs were screened in the genome sequence of *igf2*. Four SNPs located in the intron were associated significantly with growth traits, which may be useful for genetic improvement in *L. maculatus* breeding. These results are important markers for the marker-assisted selection in *L. maculatus*.

## Figures and Tables

**Figure 1 animals-13-00982-f001:**
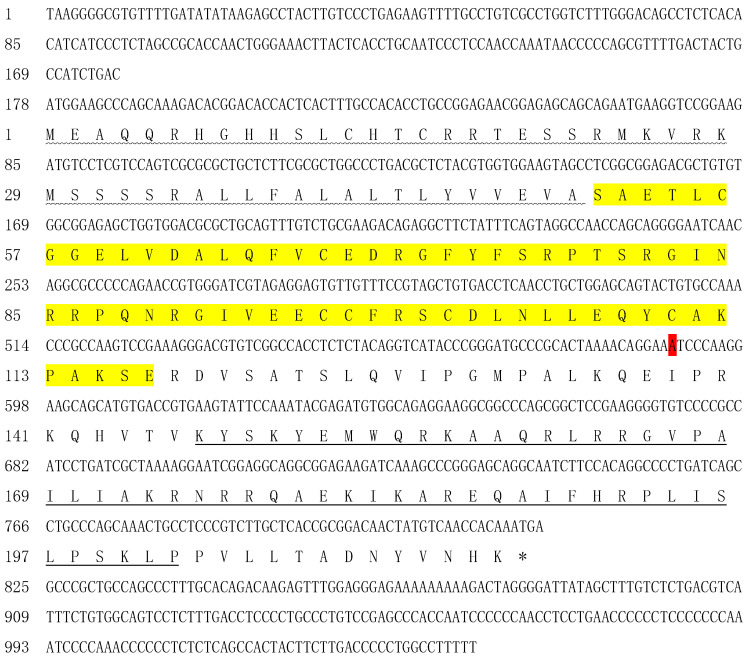
cDNA and deduced amino acid sequence of *igf2*. The positions of the cDNA and deduced amino acid sequences are shown by numbers on the left. The signal peptide is presented as a wavy line. The IGF domain is in yellow. The IGF2_C domain is underlined. The SNP is in red.

**Figure 2 animals-13-00982-f002:**
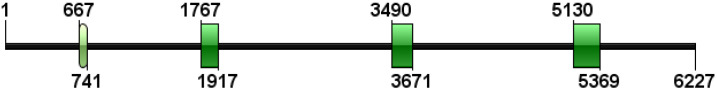
Structure of the genome sequence of *igf2*. The black straight line represents a noncoding region. Green boxes represent exons.

**Figure 3 animals-13-00982-f003:**
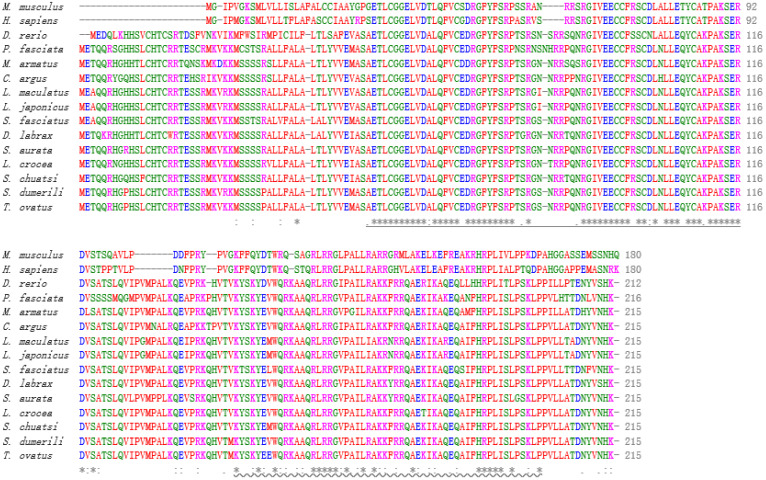
Multiple alignments of IGF2 sequences from various species. The IGF domain is shown with an underline. IGF2_C is represented with a wavy line. Single, strong, or weakly conserved residues are indicated by “*”, “:”, and “.”, respectively. The accession numbers for sequences are shown in Table 2.

**Figure 4 animals-13-00982-f004:**
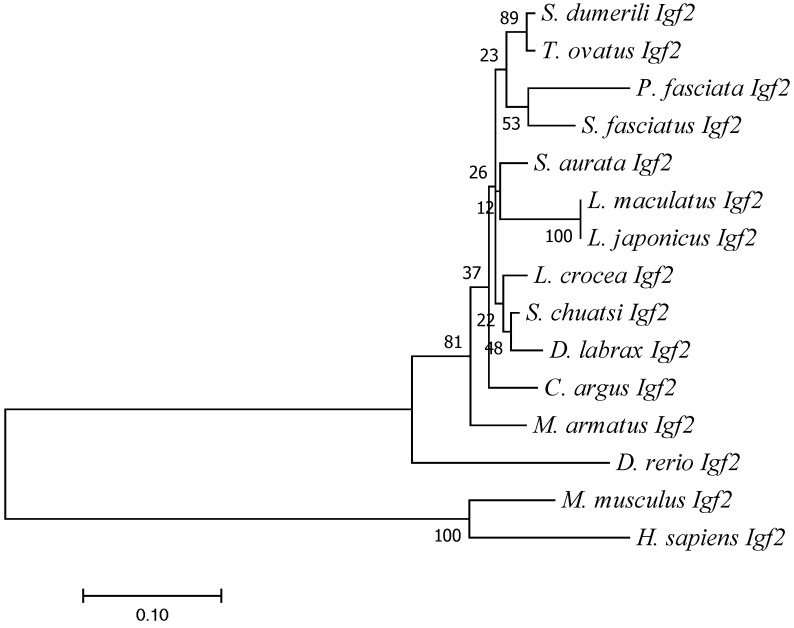
Phylogenetic tree based on the amino acid sequence of IGF2. The accession numbers for sequences are shown in Table 2.

**Figure 5 animals-13-00982-f005:**
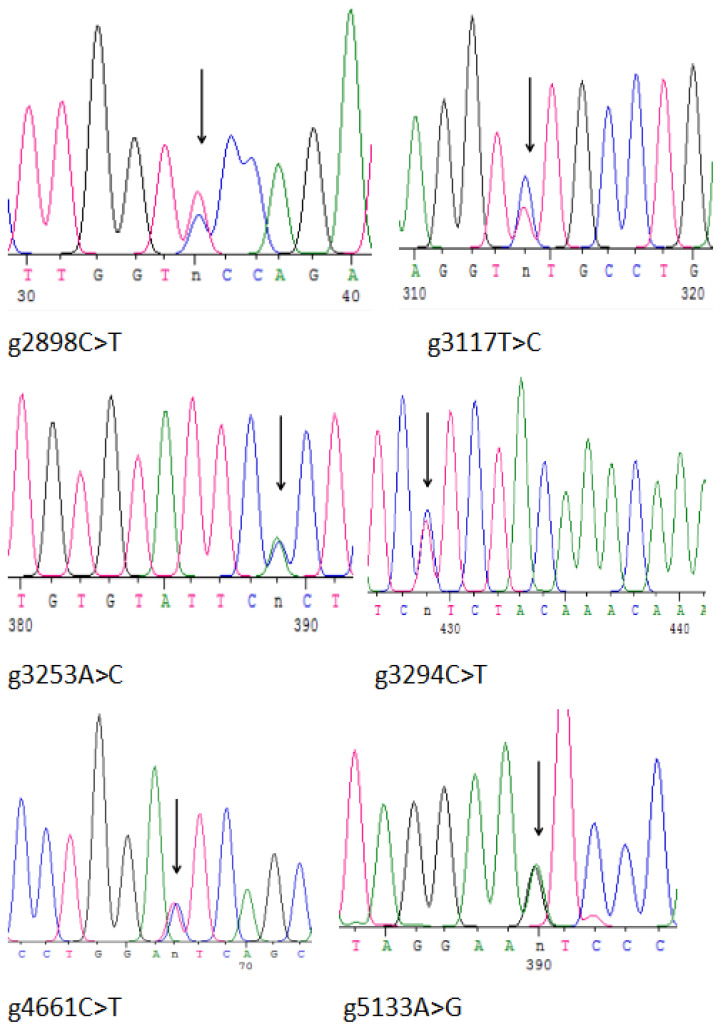
Peak chart of the SNPs of *igf2* in *L. maculatus.* The arrows represent the mutation position.

**Table 1 animals-13-00982-t001:** Primers used in this study.

Primer Name	Sequence (5’→3’)	Product Length (bp)	Purpose
cs	CCGCACTAAAACAGGAAATC	129	Verification of cDNA sequence
ca	TCCTTTTAGCGATCAGGATG		
gs	ATAAGAGCCTACTTGTCCCT	3901	Verification of DNA sequence
ga	GGCAATTCAGACAGAAGGTA		
E1s	TGAGAATGCACTTGAGTCAG	714	SNP identification at 108-821 bp
E1a	CGTTTTAACCTCACAACTGG		
E2s	CCTGTTATGAGAGGCAACAT	953	SNP identification at 1558-2510 bp
E2a	GTTGATGTTAATGATCCGCC		
E3s	GTTCCTCTACCTGCAATCAA	1036	SNP identification at 2826-3861 bp
E3a	GGAAGGTAACTGCTTACCAA		
E4s	GGGGATAAACACTCCTCAAG	1677	SNP identification ay 4542-6218 bp
E4a	CAACCTCTTCTTTGAGTCCA		

**Table 2 animals-13-00982-t002:** IGF2 sequence of each species used in this study.

Latin Name	Accession Number	Identity (%)
*Lateolabrax maculatus*	ON462263	–
*Lateolabrax japonicus*	AEX60713.2	100
*Siniperca chuatsi*	ADO14143.1	93
*Seriola dumerili*	XP_022609942.1	93
*Trachinotus ovatus*	ANJ77685.1	93
*Larimichthys crocea*	XP_019124122.1	92
*Salarias fasciatus*	XP_029951810.1	92
*Sparus aurata*	ABQ52655.1	92
*Dicentrarchus labrax*	AAW29441.1	90
*Channa argus*	KAF3689021.1	88
*Mastacembelus armatus*	XP_026167195.1	87
*Poeciliopsis fasciata*	ABC60243.1	87
*Danio rerio*	NP_001001815.1	76
*Mus musculus*	AAH53489.1	47
*Homo sapiens*	AAB34155.1	44

**Table 3 animals-13-00982-t003:** Fourteen SNP alleles and genotype frequency of *igf2*.

Locus	Position	Genotype and Number	Genotype Frequency
g2898C>T	Intron 2	CC(153)/TT(12)/TC(25)	0.81/0.06/0.13
g2907C>T	Intron 2	CC(153)/TC(37)	0.81/0.19
g3016A>C	Intron 2	AA(165)/CC(22)/AC(3)	0.92/0.11/0.02
g3117T>C	Intron 2	TT(178)/CC(2)/TC(10)	0.94/0.01/0.05
g3230A>C	Intron 2	AA(136)/CC(9)/AC(45)	0.72/0.05/0.23
g3253A>C	Intron 2	AA(111)/CC(26)/AC(53)	0.58/0.14/0.28
g3294C>T	Intron 2	TT(81)/CC(97)/TC(12)	0.43/0.51/0.06
g3329G>A	Intron 2	AA(19)/GG(171)	0.10/0.90
g3435C>A	Intron 2	AA(11)/CC(179)	0.06/0.94
g3474T>G	Intron 2	TT(175)/GG(15)	0.92/0.08
g4661C>T	Intron 3	CC(149)/TT(10)/TC(28)	0.80/0.05/0.15
g4927A>G	Intron 3	GG(23)/AA(167)	0.12/0.88
g5064C>T	Intron 3	CC(131)/TT(59)	0.69/0.31
g5133A>G	Exon 4	AA(140)/GG(26)/GA(24)	0.74/0.14/0.12

**Table 4 animals-13-00982-t004:** Association of genotypes of *igf2* SNPs with growth trait in *L. maculatus.*

Locus	Genotype	BM (g)	SL (cm)	BD (mm)	BW (mm)	HL (mm)	TL (cm)
g2907C>T	CC	586.34 ± 133.16	33.29 ± 2.31	82.52 ± 11.63 ^a^	43.08 ± 8.08	92.30 ± 9.75 ^a^	36.33 ± 2.56
	TC	594.11 ± 104.86	33.29 ± 2.01	86.90 ± 8.69 ^b^	44.56 ± 6.71	95.8181 ± 7.66 ^b^	36.72 ± 2.18
g3230A>C	AA	595.23 ± 127.35	33.50 ± 2.21	83.47 ± 12.11	43.35 ± 8.28	93.28 ± 9.77	36.67 ± 2.39 ^a^
	CC	641.84 ± 129.13	33.43 ± 1.48	86.11 ± 6.74	46.29 ± 5.35	95.12 ± 6.91	37.72 ± 2.30 ^a^
	AC	564.70 ± 131.72	32.79 ± 2.50	82.58 ± 8.89	42.87 ± 6.60	91.51 ± 8.82	35.56 ± 2.76 ^b^
g3294C>T	CC	580.76 ± 137.30	33.32 ± 2.23	81.81 ± 11.72 ^a^	41.53 ± 9.24 ^a^	91.31 ± 9.94 ^a^	36.07 ± 2.53
	TC	578.12 ± 96.09	33.26 ± 1.47	81.13 ± 13.01 ^ab^	44.02 ± 5.51 ^ab^	90.17 ± 10.10 ^ab^	36.62 ± 1.61
	TT	603.29 ± 122.90	33.36 ± 2.42	85.60 ± 10.02 ^b^	45.48 ± 5.34 ^b^	95.33 ± 8.24 ^b^	36.89 ± 2.58
g5064C>T	CC	603.03 ± 134.40	33.61 ± 2.43 ^a^	83.73 ± 10.56	43.53 ± 7.91	92.94 ± 8.80	36.61 ± 2.65
	TT	566.32 ± 117.03	32.86 ± 1.80 ^b^	82.22 ± 12.46	42.86 ± 7.95	92.49 ± 11.12	36.19 ± 2.26

Notes: Different superscript letters within a column denote a significant difference (*p < 0.05*). Values are the mean ± standard deviation. SL: standard length, BM: body mass, BD: body depth, BW: body width, HL: head length, TL: total length.

## Data Availability

Data are contained within the article.

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
