# Peer review of "Molecular Cloning, Screening of Single Nucleotide Polymorphisms, and Analysis of Growth-Associated Traits of igf2 in Spotted Sea Bass (Lateolabrax maculatus)"

_animals, 2023, doi:10.3390/ani13060982_

Round 1

Reviewer 1 Report

The present study reported the molecular characteristics of insulin-like growth factor 2 (igf2) gene and identified the single nucleotide polymorphisms (SNPs) of igf2 from the spotted sea bass (Lateolabrax maculatus), and some of these SNPs were significantly associated with growth traits. The result could be valuable in genetic breeding of L. maculatus. However, the similar work has been reported by several studies in fish or other organisms, what’s the novelty of this work?

Specific comments:

Lines 135-141: Is there any relevant reference on the model? If yes, please add the references.

Lines 247-248: change “Li et al. (2012)[23]…” with “Li et al. (2012)[22]…”

Discussion section: How to validate the isolated SNPs are indeed related to growth traits?

Author Response

Dear Reviewer,

Thank you for your comments to our manuscript. The revised manuscript and the response to you are sending to you now. Please see the attachment.

We tried our best to answer all questions point by point. All modifications were marked with red in the manuscript.

Sincerely yours

Sigang Fan

Reviewer 2 Report

This paper has merit in that it reports information that is critical for selective breeding in the selected aquaculture species and by extenstion to other related aquaculture fish species.  The core genetic data seems sound however a serious flaw in their presentation of "growth" data and analysis causes the need for major revision.  The metrics of growth are not true measures of growth without the demographic information about the fish sampled. They are simply morphometric measures.  The authors need to provide the culture history of the fish, did they come from the same group?, how many parents were used to produce the group?, what is the age of the fish? etc... without this information the postulation of the utility of the genetic markers relationship to growth is suspect. I have attached my notes as I read the paper as a PDF with comments related to edits that must be made along with this critical issue.  Specific needs that I want to highlight include:

-The metrics of growth presented in the materials and methods do not match the infromation in the citation [20] clearly and the terminology is not consistent with that paper. 

-The authors present general linear model as the method used for determination to say SNP and growth data were "..correlatively analyed...". However, correlation analysis measures the strength of the relationship between two variables with a scale of -1 to +1, while the R-squared reported by GLM measures the amount of variation in the data that is explained by the model with a scale of 0 to 1. These are not equivalent analyses.  If correlation is desired a correlation analysis such as a Pearson's correlation coefficients (r) should be conducted.  

- The authors present levels of significance for their analysis reather than a discrete alpha level.  The alpha needs to be determined a priori to meet the assumptions of the test being conducted and then the decision of significance is made based on the p value calculated for the test and the a priori alpha.  Then the measure is judged as either statistically significant or not.  There are no levels of significance (e.g., "extreamely significant" as put forth by the authors). Then, in the results the actual calculated p value for each test should be reported rather than the decision level (e.g., "P<0.05" as reported by the authors) (note: p in this case is lower case not capitalized since it refers to a particular data realization not probability in general).

- for "body length" and "whole length" are these standard length and total length respectively, the authors should use the standard terminology of fisheries professionals.  All of these metrics should be defined in this paper, if space is not a constraint, as the reader may or may not have access to the referenced paper (i.e., [20]).

-The majority of the english prose was good but there are some issues of wording and sentence structure that would benefit from a pre-review by a native english writer

Author Response

Dear Reviewer,

Thank you for your comments to our manuscript. The revised manuscript and the response to you are sending to you now. Please see the attachment.

We tried our best to answer all questions point by point. All modifications were marked up using the “Track Changes” function in the manuscript.

Sincerely yours

Sigang Fan

Reviewer 3 Report

It seems to me a study that provides important knowledge about how gene expression affects growth. Only a few small modifications are necessary.

Add the source of the SPSS software.

Throughout the text it only mentions P<0.05, however in line 141 it refers to P<0.01, perhaps it should be eliminated.

All table legend must be above according to the authors guide

Author Response

(The authors gave the same response as above.)

Reviewer 4 Report

In this manuscript the genome sequence of igf2 is isolated in the spotted sea bass, and fourteen SNPs were screened. Results show that four of them seem to be associated with growth trait in this teleost.

 -        The use of the English language in this manuscript should be extensively reviewed.

-        It is important to revise the names of genes (in lowercase and italics) and the proteins (in uppercase) throughout the manuscript.

-        Some additional information regarding fish used in this study should be provided, e.g. the sex, age, weight and size of the animals. Which is the tissue used for the RNA isolation? Which are the reference genes in the qPCR analysis?

-        Some recent references on the IGF system in fish (reviews) should be included in the manuscript. In addition, it is important to specify in the sentences whether they refer to mammals or fish, as important differences exist in the IGF axis among vertebrate classes.

-        The use of abbreviations in the abstract should be reduced to the minimum.

-        The existence of more than one paralog in fish (line 44) is the consequence of the whole genome duplication 3R experienced by teleost [Hoegg et al. J. Mol. Evol. 2004, 59, 190–203], followed by an extra round of genome duplication (4R) that occurred in both salmonid and the Cyprininae subfamily (Macquen and Johnston. Proc. R. Soc. B Biol. Sci. 2014, 281, 20132881) This explanation should be indicated.

-        Regarding the figures and tables, the legends and captions should include all the abbreviations. Figure 1 may be supplementary material. Figure 5 should be improved as it is not clear in its current format. Fig 4 is of low quality; it is difficult to identify the symbols that indicate the conserved residues.

-        Line 185 Which class of vertebrate are considered as the higher vertebrates? In this journal it is important to clearly indicate the class of vertebrates.

-        Line 187: typographic mistake, “suggested”.

-        Line 241: The sentence which indicates that the function of vertebrate IGF2 protein is highly conserved is not supported by results in this manuscript.

-        Line 247: Mus musculus, genus (Mus) in uppercase and species (musculus) in lowercase.

-        Line 245: the authors suggest that fish igf2 genome is relatively conserved. However, in line 248 it is said that the igf2 genome structure has greatly changed during evolution. These two sentences that seem to be contradictory should be explained.

Author Response

Comments and Suggestions for Authors

In this manuscript the genome sequence of igf2 is isolated in the spotted sea bass, and fourteen SNPs were screened. Results show that four of them seem to be associated with growth trait in this teleost.

 -        The use of the English language in this manuscript should be extensively reviewed.

Response:Thank you for your comments.We have checked it.

-        It is important to revise the names of genes (in lowercase and italics) and the proteins (in uppercase) throughout the manuscript.

Response:Thank you for your comments.I have checked it. As editor`s advice in November,2022. “Igf2 for the name of the protein, and igf2, in italics, for the gene.”, we have corrected it. This is followed ZFIN Zebrafish Nomenclature Conventions (ZFIN) 

-        Some additional information regarding fish used in this study should be provided, e.g. the sex, age, weight and size of the animals. Which is the tissue used for the RNA isolation? Which are the reference genes in the qPCR analysis?

Response: Thank you for your comment. All the spotted sea bass in this study are 9 month old. Sex could not be determined because it takes four years for the spotted sea bass to reach sexual maturity. The average standard length of L. maculatus individuals were 33.34 ± 2.27 cm (Line 83-84). Muscle were used for the RNA isolation. The results of qPCR was unrelated to the research purpose. The qPCR test related content in manuscript were deleted in earlier versions.

-        Some recent references on the IGF system in fish (reviews) should be included in the manuscript. In addition, it is important to specify in the sentences whether they refer to mammals or fish, as important differences exist in the IGF axis among vertebrate classes.

Response: Thank you for your comment.We have added or changed some recently references. The sentences whether they refer to mammals or fish were checked.

  1. Zeng, N., Bao, J., Shu, T., Shi, C., Zhai, G., Jin, X., He, J., Lou, Q., Yin, Z. Sexual dimorphic effects of igf1 deficiency on metabolism in zebrafish. Front endocrinol 2022, 13, 879962.
  2. Li, J., Liu, Z., Kang, T., Li, M., Wang, D., & Cheng, C. H. K. (2021). Igf3: a novel player in fish reproduction†. Biology of reproduction, 104(6), 1194–    
  3. Chen, X., Peng, L. B., Wang, D., Zhu, Q. L., & Zheng, J. L. (2022). Combined effects of polystyrene microplastics and cadmium on oxidative stress, apoptosis, and GH/IGF axis in zebrafish early life stages. The Science of the total environment, 813, 152514.

-        The use of abbreviations in the abstract should be reduced to the minimum.

Response: Thank you for your comment. I have corrected it. Line 19 SNPs; Line 22 ORF; Line 24 IIGF.

-        The existence of more than one paralog in fish (line 44) is the consequence of the whole genome duplication 3R experienced by teleost [Hoegg et al. J. Mol. Evol. 2004, 59, 190–203], followed by an extra round of genome duplication (4R) that occurred in both salmonid and the Cyprininae subfamily (Macquen and Johnston. Proc. R. Soc. B Biol. Sci. 2014, 281, 20132881) This explanation should be indicated.

Response: Thank you for your comment. I have added these sentences (Line 47-51)

Two references were added in paper.

Hoegg, S., Brinkmann, H., Taylor, J. S., & Meyer, A. (2004). Phylogenetic timing of the fish-specific genome duplication correlates with the diversification of teleost fish. Journal of molecular evolution, 59(2), 190–203. https://doi.org/10.1007/s00239-004-2613-z

Macqueen, D. J., & Johnston, I. A. (2014). A well-constrained estimate for the timing of the salmonid whole genome duplication reveals major decoupling from species diversification. Proceedings. Biological sciences, 281(1778), 20132881. https://doi.org/10.1098/rspb.2013.2881

-        Regarding the figures and tables, the legends and captions should include all the abbreviations. Figure 1 may be supplementary material. Figure 5 should be improved as it is not clear in its current format. Fig 4 is of low quality; it is difficult to identify the symbols that indicate the conserved residues.

Response: Thank you for your comment. Figure 1 corrected into Figure S1 which was provided as supplementary materia. Vector diagram of Figure 5 (corrected into Figure 4) and Figure 4 (corrected into Figure 3) were provided. It is clear.

-        Line 185 Which class of vertebrate are considered as the higher vertebrates? In this journal it is important to clearly indicate the class of vertebrates.

Response: Thank you for your comment.Mammals are higher vertebrates.I have corrected it (Line 189).

-        Line 187: typographic mistake, “suggested”.

Response:Sorry for the mistake. Corrected.

-        Line 241: The sentence which indicates that the function of vertebrate IGF2 protein is highly conserved is not supported by results in this manuscript.

Response: Thank you for your comment.This sentence was deleted.

-        Line 247: Mus musculus, genus (Mus) in uppercase and species (musculus) in lowercase.

Response: Thank you. I have corrected it.

-        Line 245: the authors suggest that fish igf2 genome is relatively conserved. However, in line 248 it is said that the igf2 genome structure has greatly changed during evolution. These two sentences that seem to be contradictory should be explained.

Response: Thank you for your comment. This is not contradictory. The structure of the fish igf2 genome is relatively conserved, because most of fish igf2 genome are included four exons and three introns. Therefore, it is relatively conserved in fish. However, the structure of mammal igf2 genome are included 8-10 exons, which is obviously different with the structure of fish igf2 genome. Therefore, from fish to mammals,the igf2 genome structure has greatly changed.

Round 2

Reviewer 2 Report

Line 12 and 13 The sentence "In order to promote the progress of L. maculatus breeding, SNPs which significantly with growth trait were screened." needs to be corrected there is a word missing between "significantly" and "with growth" I suggest "In order to promote the progress of L. maculatus breeding, SNPs which are significantly associated with growth traits were screened."

Line 29 delete the period between "breeding" and "program"

Lines 78 and 79 "The body mass (BM) was weighed with the fish scales." does this mean with the scales still attached to the skin or using a balance?

It is still unclear what the growth traints measured were.  The metrics presented are morphometric measures that are not generally used as expressions of growth and only imply growth since all of the measures were taken on fish that were 9 months old. None of theses express growth as a rate as is the standard.  They are instantaneous measures.  In the methods section clarify that the BM, SL, BD, BW, HL and TL at nine months were used as measures of growth for the analysis.

Author Response

Comments and Suggestions for Authors

Line 12 and 13 The sentence "In order to promote the progress of L. maculatus breeding, SNPs which significantly with growth trait were screened." needs to be corrected there is a word missing between "significantly" and "with growth" I suggest "In order to promote the progress of L. maculatus breeding, SNPs which are significantly associated with growth traits were screened."

Response: OK. I have corrected it.

Line 29 delete the period between "breeding" and "program"

Response: Thank you. I have corrected it.

Lines 78 and 79 "The body mass (BM) was weighed with the fish scales." does this mean with the scales still attached to the skin or using a balance?

Response: Thank you for your comment. Fish was weighed with the scale showed as followed (Figure 1). When we press the peeling button, the number of weight was become zero. Then, we put the fish on the scale. Therefore, the net weight of each fish was weighed and recorded. Each fish was cleaned before it was weighted. Few skin was attached on scale. If the scale pan was dirty, we would clean and dry it, which will not affect the measure of body mass.

Figure 1. The fish scale

It is still unclear what the growth traints measured were.  The metrics presented are morphometric measures that are not generally used as expressions of growth and only imply growth since all of the measures were taken on fish that were 9 months old. None of theses express growth as a rate as is the standard.  They are instantaneous measures.  In the methods section clarify that the BM, SL, BD, BW, HL and TL at nine months were used as measures of growth for the analysis.

Response: Thank you for your comment. A sentence was added in line 87-88. “These growth traits at nine months were measured and analyzed.”

Reviewer 4 Report

I have no additional comments

Author Response

Dear  reviewer,

Thank you very much.

Sincerely,

Sigang Fan